# Coffee House Consumers' Value Perception and Its Consequences: Multi-Dimensional Approach

**Kwangyong Kim [1], Hyun-jun Choi [2] and Sunghyup Sean Hyun [3],***

[1]  Korean Standards Association, 5, Teheran-ro 69-gil, Gangnam-gu, Seoul 06152, Korea; kimky@ksa.or.kr
[2]  Department of Hotel and Foodservice Management, Cheong Ju University, 298 Daesung-ro, Cheongwon-gu, Cheong Ju, Korea; chjun92@naver.com
[3]  School of Tourism, Hanyang University, 17 Haengdang-dong, Seongdonggu, Seoul 04763, Korea
*  Correspondence: sshyun@hanyang.ac.kr; Tel.: +82-2-2220-0862

**Abstract:** This research paper explored the causal relationship between consumers' brand value perception, brand credibility, brand prestige, and other consequential outcome variables. The conceptual relationships were described into a potential theoretical research model. The potential theoretical model was empirically assessed using structural equation-based modeling analysis using the survey data obtained from 309 coffee drinkers in United States. The data analysis outcomes revealed that utilitarian-related value perception creates consumers' brand credibility. Moreover, coffeehouse consumers' hedonic value perception and social value perception increase brand prestige perception. This study revealed that consumers' credibility perception and perceived prestige level has direct/indirect influence on perceived brand trust, commitment, and loyalty. Based upon the research findings, suggestions for the industry and future studies are provided.

**Keywords:** utilitarian value; hedonic value; social value; brand commitment; behavioral loyalty; brand credibility; brand prestige; coffeehouse

## 1. Introduction

Consumers are value-driven [1,2]. Customers seek more than cheap price and nice location in various buying situations. Maximizing consumers' perceived value is an obligatory condition for practitioners in order to survive in a highly competitive marketplace [3]. Hence, consumers' value perception has been a topic of importance in order to understand consumer buying intention in the hospitality industry [4,5]. Despite the high importance of the topic, lack of attention has been given to it in the hospitality research community [6]. Regarding these under-researched, yet critical issues, no efforts have been made to understand their role in brand-related consumer attitudes.

Previous studies discussed how brand influences consumers' behavior. For instance, Jeng (2016) [7] analyzed how airline passengers' brand credibility influences ticket purchase intentions. Expanding Jeng's research, An, Do, Ngo, and Quan (2019) [8] examined how brand credibility leads to positive word of mouth. Hwang and Lee's (2019) [9] brand research revealed that consumers' brand prestige perception can create well-being perception when they actually purchase the service/product. In 2015, Jin, Lee, and Jun [10] investigated how brand credibility has an influential power on luxury restaurant consumers' buying intentions. Expanding their research, Kim, Ham, Moon, Chua, and Han (2019) [11] showed how CROCERANT customers' brand prestige perception induces value perceptions in restaurants.

A major function of a brand is to be recognized (i.e., being included in a potential buyer's consideration set) and to be selected (i.e., resulted in an actual usage or consumption). In an established relationship (regardless of whether the relationship is shallow or deep), the purpose of

a relationship-focused view of consumption is to make products/services be used continuously, more frequently. In fact, Erdem and Swait [12] demonstrated that consumers' perceived brand credibility, with two sub-dimensions (i.e., trustworthiness and expertise), affects brand consideration and choice.

Consumers often evaluate the totality of a consumption experience rather than attribute level performance evaluation, particularly when it is hard to articulate differences among top-performing brands. Therefore, attribute specific performance would not be the main driver of consumer choice behavior. Instead, consumers may rely on what the brand delivers (how prestigious and credible it is). To be initially selected and consumed continuously, the brand has to signify (deliver) its trustworthiness and expertise (i.e., brand credibility). Recently, Baek, Kim and Yu [13] examined the effects of brand credibility level and prestige level on a potential buyer's decision process.

Despite the critical function of customers' perceived value in a marketplace, a very limited number of research projects have been carried out to understand dimension-based brand value perception (i.e., social value perception, utilitarian value perception, and hedonic value perception). Further, the contributing roles of prestige level and brand credibility level in shaping brand relation-based quality and loyalty have not been explored [13]. Finally, empirical research offering a relationship-oriented view of consumer-brand interactions has been limited [14].

Hence, considering that there is a limited amount of academic efforts in this area, it is necessary to conduct research into this subject. Therefore, the objectives of this study are: (1) examining the impacts of three perceived value perceptions (social, hedonic, and utilitarian) on brand prestige, (2) investigating the role of value perception level on brand credibility, (3) analyzing how brand credibility and prestige level creates trust in consumers' minds, and (4) showing how brand credibility, prestige, and trust leads to band loyalty. This study set the boundary into a coffeehouse in the United States. The context offers the opportunity for multiple relatively-long interactions and communications between the customer and the product/service being offered. The conclusions of this study show meaningful instruction for branding strategies and contribute to the further understanding of the consumers' value perceptions. In the following, we first briefly introduce literature on constructs, followed by our rationale for proposing relationships, debrief methodology, we summarize our findings, and finally provide our conclusions with managerial implications.

## 2. Literature Review

### 2.1. Brand Value

It is evidenced in the existing literature that brand loyalty provides businesses with marketing advantages such as reduced marketing costs and positive word of mouth communication [15–17]. Among the key constructs, perceived value received much attention from scholars. There are various definitions of the key construct of this study: perceived value. Among the definitions, Zeithmal's [18] definition is widely accepted in consumer behavior literature. Zeithmal [18] defined perceived value perception as "the consumer's subjective/objective evaluation towards the utility of a service with regards to the perceptions of what is obtained and what is provided (p. 14)." In the given definition, value indicates the relative utility. Therefore, a more inclusive conceptualization and sophisticated measure have been called upon. More importantly, scholars postulated that perceived value of consumers can be categorized as a multi-dimensional concept [2,19,20]. Later, Sheth et al. [21] suggested that perceived value is a five-dimensional construct: emotional, functional, social, conditional, and epistemic. Such multi-dimensional conceptualization provided the best foundation for a better conceptualization of perceived value [2]. Gronroos [20] reduced the dimensions into two sub-dimensions (emotional and cognitive perceived value). In 2001, Sweeney and Soutar [2] postulated that consumers' perceived values are categorized into three dimensions: emotional, social, and functional values.

Among the three value perceptions, the utilitarian value perspective emphasizes customers' value perception on functional outcomes of product-related attributes in the buying situations. When consumers' expectations are satisfied, or if a balance between quality and cost is well-balanced,

the customers feel utilitarian value. Having a subjective satisfaction from the ambience leads into enjoyment (hedonic value) [19]. This is an important reason why consumers visit a coffeehouse. Many researchers have demonstrated the importance of social value. According to Rintamaki, Kanto, Kuusela, and Spence [3], social value, a less understood dimension, has been categorized into a sub-dimensional construct which contributes to utilitarian and hedonic value [22] or into one of three dimensions [2]. Integrating the previous studies, this study proposes three dimensional constructs, because coffeehouse consumers pursue utilitarian value (price), hedonic value (mood in the coffeehouse), and social value (social function in the coffee house) [3].

## 2.2. Brand Prestige in the Formation of Brand Credibility

Brand credibility refers to "the believability of the product information contained in a product" [7]. According to them, a brand corresponds to the accumulation of past marketing strategies and activities. Consumers believe that highly credible brands will perform consistently what they promise to consumers, and therefore, less risks are associated with the consumption of such brands. Brand credibility is critical particularly when consumers are uncertain about brands, and when the approach to information is limited. A credible brand, therefore, equates to the cost effectiveness in communication with consumers as it assures high message acceptance [13].

Jeng (2016) [7] conducted an empirical study to check the impact of brand credibility on consumers' actual ticket buying intentions. They conducted a structural equation modeling method with data collected from airline passengers. They concluded that consumers' perceived credibility towards brand increases decision convenience, and thus creates perceived loyalty. An, Do, Ngo, and Quan (2019) [8] used a sample of students to reveal how brand credibility creates students' WOM spread out intentions. Their analysis revealed that brand plays a symbolic role when consumers consider actual purchase.

Hwang and Lee (2019) [9] used a sample of senior tourists to examine their brand prestige perception. SEM outputs revealed that seniors' prestige perception directly leads to well-being perception, thus deriving brand attachment. Jin, Lee, and Jun's (2015) [10] study found that luxury restaurant brand's prestige level plays a critical role in the formation of luxury restaurant selection. Their study expanded brand prestige research into the restaurant field, thus expanding the previous brand prestige research area. Their study was further expanded by Kim, Ham, Moon, Chua, and Han (2019) [11]. Kim, Ham, Moon, Chua, and Han (2019) [11] analyzed a restaurants' GROCERANT customers group, and found that restaurant experience creates prestige level. Then, the prestige level enhances value perception and loyalty perception.

## 2.3. Brand Trust

The degree of the feeling of uncertainty in service purchase and consumption creates perceived risk and aversion. This study borrowed the definition of brand trust suggested by Chaudhuri and Holbrook [11] which defines it as the willingness of the buyer relying on the brand performing its stated role. Therefore, being trusted implies that the user has high confidence in performance and a lack of uncertainty.

## 2.4. Brand Loyalty

It has been well demonstrated that loyal consumers are more willing to build a mutually beneficial relationship and favorable behavioral outcomes [17]. The role of brand in such relationships is well discussed [15]. Although brand loyalty has been discussed over three decades, it has been limitedly defined only from a behavioral perspective [23]. Dick and Basu's [17] conceptualization of loyalty encompassed the two dimensions of loyalty: attitude and behavior (more specifically favorable attitude and repeat purchase).

There are various definitions of brand loyalty. Oliver [24] suggested this definition of brand loyalty: "a subjective belief to re-purchase a favorable service continuously, consequently leading to repetitive same company's service" (p. 34).

Attitudinal loyalty indicates the commitment level that consumers have toward the brand, whereas the purchase loyalty is the willingness of consumers to purchase the brand [16,25–27]. Other researchers [28–30] suggest encompassing purchase loyalty in behavioral intention. In this study, behavioral loyalty includes not only the purchase intention but also the behavioral intention to recommend the brand to others.

### 2.5. Relationship among Consumer Value Perception, Brand Prestige, Brand Credibility, and Relational Outcomes

It is supported in the literature that the effects of value dimensions may depend on the type of expected service or product being considered [2]. Consistency in performance over time and signals of the quality of the given service/product are important attributes in deciding brand credibility.

Brand prestige indicates a brand's positioning strategy which places a relatively high level of luxury perception into consumers' minds [13,31]. Baek et al. [13] postulated that brand credibility is directly associated with tangible and utilitarian sides of value perception. They argue that brand credibility could be created via higher consistency. Therefore, utilitarian value shapes brand credibility and, on the other hand, it is expected that hedonic quality cues and social benefits will eventually enhance brand prestige.

Integrating the above discussion, this study proposes that the consumer's utilitarian value perception that patrons develop through buying and consumption of a product/service will increase consumers' perception of brand credibility. On the contrary, social and hedonic value that customers perceive will have a direct impact on brand prestige.

**Hypothesis 1 (H1):** *Utilitarian value is directly/positively associated with brand credibility.*

**Hypothesis 2 (H2):** *Hedonic value is directly/positively associated with brand prestige.*

**Hypothesis 3 (H3):** *Social value is directly/positively associated with brand prestige.*

**Hypothesis 4 (H4):** *Utilitarian value is directly/positively associated with brand trust.*

**Hypothesis 5 (H5):** *Social value is directly/positively associated with brand commitment.*

### 2.6. Impact of Brand Prestige and Credibility on the Consumers Trust into the Brand

Erdem and Swait [12] suggested that, in order to create/form credibility, it is required to achieve two components: expertise (i.e., the ability of a firm to produce/deliver what has been expected) and trustworthiness (i.e., willingness to continuously produce/deliver what has been expected). Therefore, developing consistency in satisfying consumers' needs will create consumers' confidence/trust with regard to the reliability and integrity of a brand. Consequently, buying a product/service of a trustworthy brand can ensure promised quality [13]. For this reason, scholars [12] have postulated that brand credibility and brand prestige can enhance customers' confidence in reliability and integrity towards the brand. Integrating the above theoretical discussions, this research suggests the following hypotheses:

**Hypothesis 6 (H6):** *Brand credibility is directly/positively associated with brand trust.*

**Hypothesis 7 (H7):** *Brand credibility is directly/positively associated with brand loyalty.*

**Hypothesis 8 (H8):** *Brand prestige is directly/positively associated with brand trust.*

**Hypothesis 9 (H9):** *Brand prestige is directly/positively associated with brand commitment.*

*2.7. The role of Brand Trust in the Formation of Brand Commitment and Relational Outcomes*

The theoretical and empirical relationship between trust and company performance has been supported by previous scholars [16]. Trust has been a key concept in the formation of commitment and relationship quality [32,33]. Trust in a consumer's mind creates commitment towards a brand, thus enhancing relationship quality [34]. Hence, it can be theorized that a brand, that can create trust in consumers' minds, can boost future repurchases. Based on this logic, the following hypotheses can be proposed:

**Hypothesis 10 (H10):** *Brand trust is positively related with brand commitment.*

**Hypothesis 11 (H11):** *Brand trust is positively related with brand loyalty.*

*2.8. The role of Brand Commitment in the formation of Behavioral Loyalty*

Bandyopadhyay and Martell [23] empirically tested and concluded that behavioral loyalty is positively influenced by attitudinal loyalty. As such, we propose that a committed customer (i.e., attitudinal loyalty) will exhibit favorable behaviors related to the brand.

**Hypothesis 12 (H12):** *Brand commitment is positively associated with brand loyalty.*

## 3. Methodology

*3.1. Measurement Scale*

Measurement scales from the existing literature were borrowed and revised to fit within the boundary of the coffeehouse. All the concepts used in this research were assessed by using seven-point Likert scales (ranging from 1 to 7). Before finalizing the survey questionnaire, an expert group (academic professionals) carefully proofread the survey items to check content validity. A pilot test survey was initially conducted with a convenient sample of 40 coffeehouse customers. Cronbach alpha value was checked, and all the concepts possessed higher than the conventional cutoff of 0.70 [35].

*3.2. Data Collection and Sampling*

The authors contacted an online market research company, who possess an e-mail list of consumer panels in the United States. An online-based survey was randomly emailed out to 1475 of the coffee consumers in the United States. Out of the 1475 invitations, 316 responses were collected. Seven responses were dropped because they possessed missing values. Finally, a sample of 309 was used for the actual data analysis. Thus, the usable response rate was 20.95%. Respondents were asked to insert a name of a coffeehouse that they had visited frequently, and then they answered the questionnaire based on their personal experience with that coffeehouse. A majority of the respondents were white (n = 198, 79.9%). A total of 64.1% were female (n = 247). Their age ranged from 18 to 84 years old (mean age = 44.6 years old). Regarding income level, the sample was moderately evenly distributed.

## 4. Data Analysis and Results

*4.1. CFA and Measurement Fit Estimation*

The measurement fit was estimated via structural equation modeling analysis [36]. The goodness of the measurement model fit of the CFA results was acceptable, revealing that the measurement structural model had an acceptable fit with the empirical data (IFI = 0.928, TLI = 0.918, RMSEA = 0.069, CFI = 0.927) [37]. It can be interpreted that measurement survey items are well-adapted for conducting this study. The factor loading values were ranged from 0.664 to 0.951. The factor loadings were

loaded on their matched constructs and were statistically significant at a p-value of 0.05, supporting the convergent validity of the measurement items. In addition, the average variance extracted (AVE) exceeded the cut-off value of 0.50, which confirmed convergent validity [36–38]. The squared correlation of each of the given variables was relatively smaller than the AVE of matched constructs excluding two pairs (one related to utilitarian value and the other related to social value). Following Bagozzi and Yi's [37] guidelines, the χ2 difference comparison test was conducted. The values gained from the two models (uncombined vs. combined) were statistically compared. Based on the value of chi-square comparison analysis, it was judged that the original measurement fit model is the superior one. Based on the validity check, it can be interpreted that the measurement items check what it was intended for them measure. Lastly, the composite reliability values of each concept were higher than the cut-off point value of 0.70 [39]. Table 1 presents CFA analysis results.

**Table 1.** Average variance extracted (AVE), squared correlation, and reliabilities.

|  | AVE | Utilitarian | Hedonic | Social | Credibility | Prestige | Trust | Commitment | Loyalty |
|---|---|---|---|---|---|---|---|---|---|
| Utilitarian | 0.53 | 0.82 | 0.70 | 0.76 | 0.87 | 0.72 | 0.81 | 0.65 | 0.90 |
| Hedonic | 0.62 | 0.48 | 0.89 | 0.85 | 0.65 | 0.65 | 0.66 | 0.70 | 0.68 |
| Social | 0.69 | 0.58 | 0.72 | 0.92 | 0.71 | 0.69 | 0.69 | 0.81 | 0.79 |
| Credibility | 0.85 | 0.76 | 0.43 | 0.50 | 0.97 | 0.68 | 0.91 | 0.65 | 0.82 |
| Prestige | 0.82 | 0.51 | 0.42 | 0.47 | 0.46 | 0.93 | 0.66 | 0.63 | 0.68 |
| Trust | 0.84 | 0.65 | 0.43 | 0.47 | 0.82 | 0.44 | 0.96 | 0.63 | 0.80 |
| Commitment | 0.77 | 0.42 | 0.49 | 0.66 | 0.42 | 0.39 | 0.40 | 0.93 | 0.80 |
| Loyalty | 0.74 | 0.80 | 0.47 | 0.62 | 0.67 | 0.47 | 0.63 | 0.64 | 0.94 |

*4.2. Structural Model Analysis and Testing Hypothesis*

Structural model was statistically tested, and it satisfactorily fits the collected data (RMSEA = 0.070, CFI = 0.923, IFI = 0.923, TLI = 0.916) The t-values, which are significant at the 0.05 level [30], were used for hypotheses testing. Table 2 shows hypotheses testing results (standardized coefficient value and relevant t-values presented in parenthesis).

**Table 2.** Hypotheses test.

| Hypothesis | Path | Regression Coefficient | T-value | Decision |
|---|---|---|---|---|
| 1 | Utilitarian Value→Brand Credibility | 0.900 | 12.860 | Supported |
| 2 | Hedonic Value→Brand Credibility | 0.260 | 2.360 | Supported |
| 3 | Social Value→Brand Credibility | 0.480 | 4.340 | Supported |
| 4 | Utilitarian Value→Brand Trust | - | - | Not Supported |
| 5 | Social Value→Brand Commitment | 0.720 | 11.420 | Supported |
| 6 | Brand Credibility→Brand Trust | 0.850 | 18.060 | Supported |
| 7 | Brand Credibility→Brand Loyalty | 0.540 | 11.730 | Supported |
| 8 | Brand Prestige→Brand Trust | 0.110 | 3.040 | Supported |
| 9 | Brand Prestige→Brand Commitment | - | - | Not Supported |
| 10 | Brand Trust→Brand Commitment | 0.140 | 2.500 | Supported |
| 11 | Brand Trust→Brand Loyalty | - | - | Not Supported |
| 12 | Brand Commitment → Brand Loyalty | 0.450 | 9.720 | Supported |

## 5. Conclusions

Researchers have postulated that consumers/customers' perceived value is a critical factor which creates consumers' attitudinal perception and behavioral intentions. In this sense, Pura 's [40] research revealed a causal relationship between consumer perceived value and relational outcomes. Expanding the exciting theoretical backgrounds, this study tried to deeply examine the 'reasons' by adapting multi-dimensional value perception in consumers' minds and by inserting two key constructs into the conceptual branding model.

Hypothesis 1 proposed that utilitarian value is a driver of credibility in consumers' minds. This link was supported (0.900, t = 12.860). Thus, it can be interpreted that consumer's tangible/utilitarian value perceptions (measured by taste, prompt service delivery, variety offers, and purchase convenience in the coffeehouse setting) can maximize the consumers' perceived brand credibility.

Hypothesis 2 theoretically suggested that hedonic value is a driver of prestige perception in customers' minds. This path was empirically supported (0.260, t = 2.360). It can be interpreted that consumers' subjective feelings/experiences in the coffee shop play a critical role in the prestige brand formation.

Data analysis also supported hypothesis 3 (0.480, t = 4.340). One thing that should be noted is: among the three (social, utilitarian and hedonic) values perceptions, social value had a strongest power. This finding suggests the importance of social value creation in the coffee shop management, and also in brand management in the hospitality industry.

Data analysis supported hypothesis 5 (0.720, t = 11.420). This output suggests that social value perception is a key determinant of brand commitment enhancement. In other words, it should be emphasized that, in order to create a loyal customer group, social value should be created in the coffee shop [41,42].

Regarding the consequences of brand perception, Hypothesis 6 (0.850, t = 18.060), Hypothesis 7 (0.540, t = 11.730), and Hypothesis 8 (0.110, t = 3.040) were all supported. Based on a data analysis results, this study found that consumers' brand perception is not limited to the functional level evaluation (such as service speed, coffee quality, taste, variety, convenience, etc.). Consumers' brand perception is also highly associated with physical attributes such as enjoyable stuff, and aesthetics (i.e., facility, decorations, etc.), and also associated with the socialization attributes. Therefore, coffeehouse companies should seek to find relevant strategies. For instance, in order to maximize utilitarian value, coffee taste/service should be improved More importantly, a coffeehouse brand should invest more assets to promote the social meaning of the place. It is necessary to make the consumers feel a sense of belonging with other social people, and creating social meaning would make them to feel congruent with the coffee brand.

As this study found that brand credibility is built through providing reliable service transactions, brand credibility is critical in tangiblizing the intangible service. Loyalty is a critical factor for the success of a company [43,44]. The contribution of this research from the perspective of theory development is that this study provided further theoretical support for the effects of value on brand credibility, prestige, and loyalty.

## 6. Limitation and Future Research

The existing literature (e.g., Baek et al. [13]) used a signaling theory to examine how a credible and prestigious brand serves as a signal to perceived quality, information costs saved, and other outcome constructs. On the contrary, this study examined how perceived values contribute to the development of a coffee drinker's brand credibility level and prestige perception over time. For this reason, in this study, consumers were asked to select only one of the coffee shop companies that they like. It would be very interesting to examine how coffee drinkers' credibility and perceived prestige may affect their attitudinal and behavioral intentions in a brand extension situation.

As demonstrated in the study by Baek at al. [13], the level of product/service categories in a self-expressive continuum may function differently in consumer behavioral intention formation. Different hospitality brands can be employed in the future study as the level of self-expression in the foodservice industry varies to that of the lodging industry.

Lastly, the relationship between brand trust and utilitarian value, the relationship between brand prestige and brand commitment, and the relationship between brand trust and brand loyalty were not supported in this study. Therefore, it might be a meaningful trial to re-test the relationships using different a sample in a different culture.

**Author Contributions:** Writing, review and editing, revision, project administration, resources, conceptualization—K.K.; Revision, formal analysis, visualization, validation—H.j.-C.; Supervision and research design—S.S.H. All authors have read and agreed to the published version of the manuscript.

**Funding:** This research received no external funding.

**Conflicts of Interest:** The authors declare no conflict of interest.

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
