# Peer review of "Coffee House Consumers’ Value Perception and Its Consequences: Multi-Dimensional Approach"

_sustainability, doi:10.3390/su12041663_

Round 1

Reviewer 1 Report

The study it is useful from marketing point of view and consumer orientation on market. 

Author can improve:

Introduction with more details in the field of brand and how they influence the consumers behavior  Review literature it is a little bit old must to be refresh with new  The model it is interesting but maybe more explanation for readers without knowledge in statistic. Conclusions more details and the feedback of their research.

Author Response

Reviewer #1’s comments

Introduction with more details in the field of brand and how they influence the consumers behavior.

Response

Following the reviewer’ suggestion, we have newly inserted a paragraph into the introduction section. The newly inserted paragraph shows how brand influences the consumers’ behavior. Specifically, the following paragraph has been newly inserted into introduction section:

Previous studies discussed how brand influences consumers’ behavior. For instance, Jeng (2016) [7] analyzed how airline passengers’ brand credibility influences ticket purchase intentions. Expanding Jeng’s research, An, Do, Ngo, and Quan (2019) [8] examined how brand credibility leads to positive word of mouth. Hwang and Lee (2019)’s [9] brand research revealed that consumers’ brand prestige perception can create well-being perception when they actually purchase the service/product. In 2015, Jin, Lee, and Jun [10] investigated how brand credibility has an influential power on luxury restaurant consumers’ buying intentions. Expanding their research, Kim, Ham, Moon, Chua, and Han (2019) [11] showed how CROCERANT customers’ brand prestige perception induce value perceptions in restaurants.

Newly Inserted REFERENCE

An, J., Do, D.K.X., Ngo, L.V., & Quan, T.H.M. (2019). Turning brand credibility into positive word-of-mouth: integrating the signaling and social identity perspectives. Journal of Brand Management, 26(2), 157-175.

Hwang, J. & Lee, J.H. (2019). Antecedents and consequences of brand prestige of package tour in the senior tourism industry. Asia Pacific Journal of Tourism Research, 24(7), 679-695.

Kim, S., Ham, S., Moon, H., Chua, B., & Han, H. (2019). Experience, brand prestige, perceived value (functional, hedonic, social, and financial), and loyalty among GROCERANT customers. International Journal of Hospitality Management, 77(January), 169-177.

Jeng, S.P. (2016). The influences of airline brand credibility on consumer purchase intentions. Journal of Air Transport Management. 55(August), 1-8.

Jin, J.P., Lee, S., & Jun, J. (2015). The role of brand credibility in predicting consumers' behavioural intentions in luxury restaurants. Anatolia, 26(3), 384-396.

Review literature it is a little bit old must to be refresh with new 

Response

Following the reviewer’s suggestion, we have conducted another round of literature review focused on the literature which were published recently. Especially, publications after 2015 year were newly cited. Specifically, the below recent references were newly added into review of literature section:

Jeng (2016)[7] conducted an empirical study to check the impact of brand credibility on consumers’ actual ticket buying intentions. They conducted structural equation modeling method with data collected from airline passengers. They concluded that consumers’ perceived credibility towards brand increases decision convenience, thus creative perceived loyalty. An, Do, Ngo, and Quan (2019)[8] used student sample to reveal how brand credibility create students’ WOM spread out intentions. Their analysis revealed that brand plays a symbolic role when consumers consider actual purchase.

Hwang and Lee (2019)[9] used senior tourists sample to examine their brand prestige perception. SEM outputs revealed that seniors’ prestige perception directly leads to well-being perception, thus deriving brand attachment. Jin, Lee, and Jun (2015)’s[10] study found that luxury restaurant brand’s prestige level plays a critical role in the formation of luxury restaurant selection. Their study expanded brand prestige research into restaurant field, thus expanding the previous brand prestige research area. Their study was further expanded by Kim, Ham, Moon, Chua, and Han (2019)[11]. Kim, Ham, Moon, Chua, and Han (2019)[11] analyzed restaurants’ GROCERANT customers group, and found that restaurant experience creates prestige level. Then, the prestige level enhances value perception and loyalty perception.

The model it is interesting but maybe more explanation for readers without knowledge in statistic.

Response

Following the reviewer’s suggestion, we have newly inserted a lot detained explanations into the “Data Analysis and Results” section. For the readers who do not have statistical knowledge, a possible interpretations were newly added into the section.  

Conclusions more details and the feedback of their research.

Response

Following the reviewer’s suggestion, the conclusion section has been totally re-written. First, the research results were re-organized and re-written, so the conclusion can be correspond directly with the 12 hypotheses proposed. Second, we have newly created table 3. The new table (table 3) further explains more details of data analysis results. So, readers can easily understand the output. Specifically, the conclusion section has been re-written like below:

CONCLUSION

Researchers have postulated that consumers/customers’ perceived value is a critical factor which creates consumers’ attitudinal perception and behavioral intentions. In this sense, Pura 's [43] research revealed a causal relationship between consumer perceived value and relational outcomes. Expanding the exiting theoretical backgrounds, this study tried to deeply examine the ‘reasons’ by adapting multi-dimensional value perception in consumers’ mind and by inserting two key constructs into the conceptual branding model.

Hypothesis 1 proposed that utilitarian value is a driver of credibility in consumers’ mind. This link was supported (0.900, t=12.860). Thus, it can be interpreted that consumer’s tangible/utilitarian value perception (measured by taste, prompt service delivery, variety offers, and purchase convenience in the coffeehouse setting) can maximize the consumers’ perceived brand credibility.

Hypothesis 2 theoretically suggested that hedonic value is a driver of prestige perception in customers’ mind. This path was empirically supported (0.260, t=2.360). It can be interpreted that consumers’ subjective feeling/experience in the coffee shop plays a critical role in the prestige brand formation.

Data analysis also supported hypothesis 3 (0.480, t=4.340). One thing that should be noted is: among the three (social, utilitarian and hedonic) values perceptions, social value had a strongest power. This finding suggest the important of social value creation in the coffee shop management, and also in the brand management in the hospitality industry.

Data analysis supported hypothesis 5 (0.720, t=11.420). This output suggests that social value perception is key determinant of brand commitment enhancement. In other words, it should be emphasized that, in order to crate loyal customer group, social value should be created in the coffee shop [44, 45].

Regarding the consequences of brand perception, Hypothesis 6 (0.850, t=18.060), Hypothesis 7 (0.540, t=11.730), Hypothesis 8 (0.110, t=3.040) were all supported. Based on a data analysis results, this study found that consumers’ brand perception is not limited to the functional level evaluation (such as service speed, coffee quality, taste, variety, convenience, etc.). Consumers’ brand perception is also highly associated with physical attributes, such as enjoyment stuff, and aesthetics (i.e., facility, decorations, etc.), and also associated with the socialization attributes. Therefore, coffeehouse companies should find relevant strategies. For instance, in order to maximize utilitarian value, coffee taste/service should be improved More importantly, coffeehouse brand should invest more assets to promote social meaning of the place. It is necessary to make the consumers feel ‘belonging with other social people’ and ‘creating social meaning’ would make them to feel congruent with the coffee brand.

As this study found that brand credibility is built through providing reliable service transactions, brand credibility is critical in tangiblizing the intangible service. Loyalty is a critical factor for the success of a company [46, 47]. The contribution of this research from the perspective of theory development is that this study provided further theoretical support to the effects of value on brand credibility, prestige, and loyalty.

Reviewer 2 Report

I recommend reinforcing the objetives of the paper.

Similarly, I recommend that the author does a better connection between the background and the results of the research.

Author Response

Reviewer #2’s comments

I recommend reinforcing the objetives of the paper.

Response

Following the reviewer’s suggestion, we have newly inserted the clear objectives of this study like below:

Therefore, the objectives of this are: (1) examining the impacts of three perceived value perceptions (social, hedonic and utilitarian) on brand prestige, (2) investigating the role of value perception level on brand credibility, (3) analyzing how brand credibility and prestige level creates trust in consumers’ mind, and (4) showing how brand credibility, prestige, and trust leads to band loyalty. This study set the boundary into coffeehouse in United States.

Similarly, I recommend that the author does a better connection between the background and the results of the research

Response

Following the reviewer’s suggestion, the conclusion section has been totally re-written. First, the research results were re-organized and re-written, so the conclusion can be correspond directly with the 12 hypotheses proposed. Second, we have newly created table 3. The new table (table 3) further explains more details of data analysis results. So, readers can easily understand the output. Specifically, the conclusion section has been re-written like below:

CONCLUSION

Researchers have postulated that consumers/customers’ perceived value is a critical factor which creates consumers’ attitudinal perception and behavioral intentions. In this sense, Pura 's [43] research revealed a causal relationship between consumer perceived value and relational outcomes. Expanding the exiting theoretical backgrounds, this study tried to deeply examine the ‘reasons’ by adapting multi-dimensional value perception in consumers’ mind and by inserting two key constructs into the conceptual branding model.

Hypothesis 1 proposed that utilitarian value is a driver of credibility in consumers’ mind. This link was supported (0.900, t=12.860). Thus, it can be interpreted that consumer’s tangible/utilitarian value perception (measured by taste, prompt service delivery, variety offers, and purchase convenience in the coffeehouse setting) can maximize the consumers’ perceived brand credibility.

Hypothesis 2 theoretically suggested that hedonic value is a driver of prestige perception in customers’ mind. This path was empirically supported (0.260, t=2.360). It can be interpreted that consumers’ subjective feeling/experience in the coffee shop plays a critical role in the prestige brand formation.

Data analysis also supported hypothesis 3 (0.480, t=4.340). One thing that should be noted is: among the three (social, utilitarian and hedonic) values perceptions, social value had a strongest power. This finding suggest the important of social value creation in the coffee shop management, and also in the brand management in the hospitality industry.

Data analysis supported hypothesis 5 (0.720, t=11.420). This output suggests that social value perception is key determinant of brand commitment enhancement. In other words, it should be emphasized that, in order to crate loyal customer group, social value should be created in the coffee shop [44, 45].

Regarding the consequences of brand perception, Hypothesis 6 (0.850, t=18.060), Hypothesis 7 (0.540, t=11.730), Hypothesis 8 (0.110, t=3.040) were all supported. Based on a data analysis results, this study found that consumers’ brand perception is not limited to the functional level evaluation (such as service speed, coffee quality, taste, variety, convenience, etc.). Consumers’ brand perception is also highly associated with physical attributes, such as enjoyment stuff, and aesthetics (i.e., facility, decorations, etc.), and also associated with the socialization attributes. Therefore, coffeehouse companies should find relevant strategies. For instance, in order to maximize utilitarian value, coffee taste/service should be improved More importantly, coffeehouse brand should invest more assets to promote social meaning of the place. It is necessary to make the consumers feel ‘belonging with other social people’ and ‘creating social meaning’ would make them to feel congruent with the coffee brand.

As this study found that brand credibility is built through providing reliable service transactions, brand credibility is critical in tangiblizing the intangible service. Loyalty is a critical factor for the success of a company [46, 47]. The contribution of this research from the perspective of theory development is that this study provided further theoretical support to the effects of value on brand credibility, prestige, and loyalty.

Table 3 shows hypotheses testing results (standardized coefficient value and relevant t-values presented in parenthesis).

Table 3. Hypotheses Test

Hypothesis

Path

Regression Coefficient

T-value

Decision

1

Utilitarian Value->Brand Credibility

.900

12.860

Supported

2

Hedonic Value->Brand Credibility

.260

2.360

Supported

3

Social Value->Brand Credibility

.480

4.340

Supported

4

Utilitarian Value->Brand Trust

.007

.130

Not Supported

5

Social Value->Brand Commitment

.720

11.420

Supported

6

Brand Credibility->Brand Trust

.850

18.060

Supported

7

Brand Credibility->Brand Loyalty

0.540

11.730

Supported

8

Brand Prestige->Brand Trust

0.110

3.040

Supported

9

Brand Prestige->Brand Commitment

0.024

0.171

Not Supported

10

Brand Trust->Brand Commitment

0.140

2.500

Supported

11

Brand Trust->Brand Loyalty

0.010

0.122

Not Supported

12

Brand Commitment-> Brand Loyalty

0.450

9.720

Supported

Reviewer 3 Report

An interesting research case was presented.The results of the study are original.

Whole article, however, requires significant improvement:

the puropse of the article should be clearly stated. research results should coresponde directly with the 12 hypotheses proposed. The presented stuctural model (4.2.) should be related to the established hypotheses. the model requires a more detalies description. the conclusion is too laconic. Statements should corespond directly to the results of the work. It is advisable to broaden litearture studies, concerning mainly on brand management issues, including brand structure.

Author Response

the puropse of the article should be clearly stated.

Response

Following the reviewer’s suggestion, we have newly inserted the clear objectives of this study like below:

Therefore, the objectives of this are: (1) examining the impacts of three perceived value perceptions (social, hedonic and utilitarian) on brand prestige, (2) investigating the role of value perception level on brand credibility, (3) analyzing how brand credibility and prestige level creates trust in consumers’ mind, and (4) showing how brand credibility, prestige, and trust leads to band loyalty. This study set the boundary into coffeehouse in United States.

research results should coresponde directly with the 12 hypotheses proposed. The presented stuctural model (4.2.) should be related to the established hypotheses. the model requires a more detalies description. the conclusion is too laconic. Statements should corespond directly to the results of the work.

Response

Following the reviewer’s suggestion, the conclusion section has been totally re-written. First, the research results were re-organized and re-written, so the conclusion can be correspond directly with the 12 hypotheses proposed. Second, we have newly created table 3. The new table (table 3) further explains more details of data analysis results. So, readers can easily understand the output. Specifically, the conclusion section has been re-written like below:

CONCLUSION

Researchers have postulated that consumers/customers’ perceived value is a critical factor which creates consumers’ attitudinal perception and behavioral intentions. In this sense, Pura 's [43] research revealed a causal relationship between consumer perceived value and relational outcomes. Expanding the exiting theoretical backgrounds, this study tried to deeply examine the ‘reasons’ by adapting multi-dimensional value perception in consumers’ mind and by inserting two key constructs into the conceptual branding model.

Hypothesis 1 proposed that utilitarian value is a driver of credibility in consumers’ mind. This link was supported (0.900, t=12.860). Thus, it can be interpreted that consumer’s tangible/utilitarian value perception (measured by taste, prompt service delivery, variety offers, and purchase convenience in the coffeehouse setting) can maximize the consumers’ perceived brand credibility.

Hypothesis 2 theoretically suggested that hedonic value is a driver of prestige perception in customers’ mind. This path was empirically supported (0.260, t=2.360). It can be interpreted that consumers’ subjective feeling/experience in the coffee shop plays a critical role in the prestige brand formation.

Data analysis also supported hypothesis 3 (0.480, t=4.340). One thing that should be noted is: among the three (social, utilitarian and hedonic) values perceptions, social value had a strongest power. This finding suggest the important of social value creation in the coffee shop management, and also in the brand management in the hospitality industry.

Data analysis supported hypothesis 5 (0.720, t=11.420). This output suggests that social value perception is key determinant of brand commitment enhancement. In other words, it should be emphasized that, in order to crate loyal customer group, social value should be created in the coffee shop [44, 45].

Regarding the consequences of brand perception, Hypothesis 6 (0.850, t=18.060), Hypothesis 7 (0.540, t=11.730), Hypothesis 8 (0.110, t=3.040) were all supported. Based on a data analysis results, this study found that consumers’ brand perception is not limited to the functional level evaluation (such as service speed, coffee quality, taste, variety, convenience, etc.). Consumers’ brand perception is also highly associated with physical attributes, such as enjoyment stuff, and aesthetics (i.e., facility, decorations, etc.), and also associated with the socialization attributes. Therefore, coffeehouse companies should find relevant strategies. For instance, in order to maximize utilitarian value, coffee taste/service should be improved More importantly, coffeehouse brand should invest more assets to promote social meaning of the place. It is necessary to make the consumers feel ‘belonging with other social people’ and ‘creating social meaning’ would make them to feel congruent with the coffee brand.

As this study found that brand credibility is built through providing reliable service transactions, brand credibility is critical in tangiblizing the intangible service. Loyalty is a critical factor for the success of a company [46, 47]. The contribution of this research from the perspective of theory development is that this study provided further theoretical support to the effects of value on brand credibility, prestige, and loyalty.

It is advisable to broaden litearture studies, concerning mainly on brand management issues, including brand structure.

Response

Following the reviewer’s suggestion, we have conducted another round of literature review, and thus broaden literature review section. Especially, publications focused on brand management and structure were newly cited. Specifically, the below paragraph was been newly added into review of literature section:

Jeng (2016)[7] conducted an empirical study to check the impact of brand credibility on consumers’ actual ticket buying intentions. They conducted structural equation modeling method with data collected from airline passengers. They concluded that consumers’ perceived credibility towards brand increases decision convenience, thus creative perceived loyalty. An, Do, Ngo, and Quan (2019)[8] used student sample to reveal how brand credibility create students’ WOM spread out intentions. Their analysis revealed that brand plays a symbolic role when consumers consider actual purchase.

Hwang and Lee (2019)[9] used senior tourists sample to examine their brand prestige perception. SEM outputs revealed that seniors’ prestige perception directly leads to well-being perception, thus deriving brand attachment. Jin, Lee, and Jun (2015)’s[10] study found that luxury restaurant brand’s prestige level plays a critical role in the formation of luxury restaurant selection. Their study expanded brand prestige research into restaurant field, thus expanding the previous brand prestige research area. Their study was further expanded by Kim, Ham, Moon, Chua, and Han (2019)[11]. Kim, Ham, Moon, Chua, and Han (2019)[11] analyzed restaurants’ GROCERANT customers group, and found that restaurant experience creates prestige level. Then, the prestige level enhances value perception and loyalty perception.

Newly Inserted REFERENCE

An, J., Do, D.K.X., Ngo, L.V., & Quan, T.H.M. (2019). Turning brand credibility into positive word-of-mouth: integrating the signaling and social identity perspectives. Journal of Brand Management, 26(2), 157-175.

Hwang, J. & Lee, J.H. (2019). Antecedents and consequences of brand prestige of package tour in the senior tourism industry. Asia Pacific Journal of Tourism Research, 24(7), 679-695.

Kim, S., Ham, S., Moon, H., Chua, B., & Han, H. (2019). Experience, brand prestige, perceived value (functional, hedonic, social, and financial), and loyalty among GROCERANT customers. International Journal of Hospitality Management, 77(January), 169-177.

Jeng, S.P. (2016). The influences of airline brand credibility on consumer purchase intentions. Journal of Air Transport Management. 55(August), 1-8.

Jin, J.P., Lee, S., & Jun, J. (2015). The role of brand credibility in predicting consumers' behavioural intentions in luxury restaurants. Anatolia, 26(3), 384-396.

Round 2

Reviewer 1 Report

The authors improve the paper.

As an opinion  maybe will be good if they put the references from Conclusion and Limitation in the Discussion part, taking in consideration the matrix of academic studies.

Also they can express their opinion about the not supported connection utilitarian value-trust-brand a future vision.

Author Response

As an opinion  maybe will be good if they put the references from Conclusion and Limitation in the Discussion part, taking in consideration the matrix of academic studies. Also they can express their opinion about the not supported connection utilitarian value-trust-brand a future vision

Response

Following the reviewer’ suggestion, we have put the references. One more thing, following the reviewer’s second suggestion, we have newly inserted our opinion about not supported hypotheses. Specifically, the following paragraph has been newly inserted:

Lastly, the relationship between brand trust and utilitarian value, the relationship between brand prestige and brand commitment, and the relationship between brand trust and brand loyalty were not supported in this study. So, it might be a meaningful trial to re-test the relationships using different sample in different culture.

Reviewer 2 Report

Congratulations

Author Response

Congratulations

Response

Thank you.

Reviewer 3 Report

I accept the Authors explanations. I make no further comments.

Author Response

I accept the Authors explanations. I make no further comments.

Response

Thank you.

Round 3

Reviewer 1 Report

The authors improve the paper.